# Is it Worth Combining *Solidago virgaurea* Extract and Antibiotics against Uropathogenic *Escherichia coli* rods? An In Vitro Model Study

**DOI:** 10.3390/pharmaceutics13040573

**Published:** 2021-04-17

**Authors:** Dorota Wojnicz, Dorota Tichaczek-Goska, Michał Gleńsk, Andrzej B. Hendrich

**Affiliations:** 1Department of Biology and Medical Parasitology, Wrocław Medical University, 50-367 Wrocław, Poland; dorota.wojnicz@umed.wroc.pl (D.W.); andrzej.hendrich@umed.wroc.pl (A.B.H.); 2Department of Pharmacognosy and Herbal Medicines, Wrocław Medical University, 50-367 Wrocław, Poland; michal.glensk@umed.wroc.pl

**Keywords:** *Solidago virgaurea* extract, amikacin, ciprofloxacin, biofilm, uropathogenic *Escherichia coli*

## Abstract

European goldenrod (*Solidago virgaurea* L.) has long been applied in traditional medicine and recommended in the prophylaxis of urinary tract infections (UTIs). However, research describing the antibacterial properties of goldenrod is very limited. Therefore, the aim of the study was to determine the effect of *S. virgaurea* extract on the survival and biofilm formation of uropathogenic *Escherichia coli*. The interactions between the goldenrod extract and antibiotics used in UTIs were established. The influence of the extract on the duration of the post-antibiotic effects (PAE) and post-antibiotic sub-MIC effects (PASME) of amikacin and ciprofloxacin were determined. Extract composition was analyzed using coupled UHPLC/MS and the spectrophotometric method. The survival of bacteria was established using the serial dilution assay. The crystal violet assay for biofilm quantification was also used. PAE and PASME were investigated using the viable count method. The obtained results indicate that *S. virgaurea* extract limits the survival of planktonic forms of bacteria and reduces 24-h biofilm. However, the combination of *S. virgaurea* extract with antibiotics weakens their antibacterial activity and shortens the duration of PAE and PASME. Therefore, when deciding to use a combination of *S. virgaurea* extract and amikacin/ciprofloxacin, it is necessary to take into account their antagonistic activity.

## 1. Introduction

European goldenrod (*Solidago virgaurea* L.) is a perennial plant species that belongs to the aster family (Asteraceae). Because of its low growth requirements, it is common in Europe, North America and Asia. From the perspective of pharmacognosy and herbal medicine, upper parts of its shoot are of great importance. The most active substances accumulate in the flowers and leaves, which gives these parts of the plant a great pharmacognostic value [1]. Extracts of *S. virgaurea* contain a wide variety of compounds such as glycosides (virgaureoside, leiocarposide) and aglycones (vanillic acid, gallic acid) [1,2,3,4,5]; polyphenolic acids (caffeic, chlorogenic, ferulic, synapic, 3-hydroxyphenylacetic, 3,4-dihydroxyphenylacetic, homovanillic) [3,5,6]; flavonoids (quercetin and kaempferol glycosides, free aglycons and cyanidin derivatives) [1,2,5,7,8,9,10]; oleanane-type triterpene saponins [5,11,12], essential oils containing monoterpenes (myrcene, limonene, sabinene) [13,14,15]; and sesquiterpenes, clerodane diterpenes, polysaccharides and polyacetylenes [16,17,18].

*S. virgaurea* has long been applied in traditional medicine. It effectively supports the treatment of vascular, kidney and digestive system diseases. In particular, goldenrod honey supports immunity and reduces infection. Goldenrod shows diuretic, anti-inflammatory and antibacterial effects. Goldenrod tea is used as a supportive remedy in poisoning and it helps to cleanse the digestive tract. People drink it to treat diarrhoea, fever and pain of varying etiology. The beneficial effects of goldenrod in the treatment of arthritis, cystitis, renal calculi and rheumatism have also been demonstrated [19]. Due to its diuretic activities, the goldenrod herb is recommended for people suffering from bacterial infections of the urinary system and kidney inflammation [20].

Urinary tract infections (UTIs) belong to the group of the most frequently mentioned infectious diseases in the human population. The main etiological agent of UTIs, both in hospital-acquired and non-hospital-acquired infections, are uropathogenic *Escherichia coli* (UPEC) strains. They possess virulence factors that enable the invasion of the urothelium and cause pathological lesions. In many cases, in spite of antimicrobial treatment, subsequent chronic and recurring UTIs occur. This may be attributable to the ineffectiveness of earlier therapy and the survival of pathogens in the urinary tract in the biofilm form [21]. Treatment failure is associated with the presence of a polysaccharide matrix surrounding the bacteria growing in the biofilm, which significantly reduces the penetration of its structure by antibiotics. Moreover, pathogens living in the structure of the biofilm show a reduced level of metabolism, which makes it easier for them to survive during antibiotic therapy, as most antibiotics act on actively dividing cells [22].

Although the etiology of UTIs itself has not changed significantly for years, strains with increasing drug-resistance are being isolated more frequently, which is a very serious clinical problem faced by today’s medicine. This phenomenon results from a continuous (frequently excessive) increase in the use of antibiotics, which results in the formation of drug resistance, especially among clinical strains. For the above reasons, the role of natural medicinal plant materials is increasingly being studied in order to reduce this negative phenomenon [23]. The combination of plant extracts and essential oils with antibiotics provides higher biological activity. One antimicrobial agent enhances the activity of the other, and finally they may act together more effectively against drug-resistant bacteria. Some researchers suggest combining antibiotics with plant compounds to either strike multiple targets in the bacterial genome or enhance the potency of the antibiotic by weakening the bacteria’s ability to develop resistance [24,25,26,27].

In the treatment of UTIs, drugs with a broad antibacterial spectrum, such as ciprofloxacin and amikacin, are used, which differ in terms of their mode of antimicrobial action. The former blocks the replication process and the latter inhibits the synthesis of bacterial proteins [28]. Unfortunately, their wide use, both for medical and non-medical purposes significantly increases the pool of antibiotics present in the environment, thus contributing to the growth of antibiotic resistance. For this reason, preparations of natural origin that could support the bactericidal effect of antibiotics are being sought. The goldenrod herb is one of the many herbal preparations recommended as an adjuvant in the treatment of UTIs. However, research describing the antibacterial properties of goldenrod is very limited. Authors have focused only on determining the minimum bactericidal concentration values or growth inhibition zones of bacteria [29,30,31]. There are no studies concerning the survival or biofilm formation by bacteria treated with *S. virgaurea* extracts. There is also no research describing the antibacterial activities of *S. virgaurea* extracts in combination with antibiotics. Therefore, the aims of our pioneering study were (i) to determine the effect of *S. virgaurea* herb extract on the survival of planktonic forms and biofilm formation by UPEC; (ii) to determine the type of interaction between goldenrod extract and the antibiotics amikacin and ciprofloxacin; and (iii) to evaluate the influence of *S. virgaurea* extract on the duration of the post-antibiotic effect (PAE) and post-antibiotic sub-MIC effect (PASME) of amikacin and ciprofloxacin.

## 2. Materials and Methods

### 2.1. Bacterial Strain

The bacterial strain used in this study was standard uropathogenic *E. coli* CFT073 (ATCC 700928) from the American Type Culture Collection characterized as being able to perform biofilm synthesis. The strain was stored on slopes containing nutrient broth and glycerol at −20 °C.

### 2.2. Antimicrobial Agents

Two antimicrobial agents used in the treatment of UTIs were used in the study: ciprofloxacin lactate (Proxacin^®^, Polfa S.A., Warsaw, Poland) and amikacin disulfate salt (Biodacyna^®^, BIOTON S.A., Warsaw, Poland).

### 2.3. Plant Material

*Solidaginis virgaureae herba* was purchased from the herbs confectioning factory “FLOS”, general partnership (Mokrsko, Poland), with marketing authorization number R/1957. Dried plant material (50 g) was ground and extracted with MeOH (0.5 L) for 24 h at room temperature. The methanol extract was concentrated to dry residue. For UHPLC analysis, the dried methanol extract was re-dissolved in MS-grade MeOH to a 1 mg/mL concentration. One milliliter of the sample was filtered using 0.22 μm PTFE single-use syringe filters (Merck-Millipore, Darmstadt, Germany), and stored at 4 °C before analysis.

### 2.4. UHPLC-DAD-ESI-MS Analysis

The Thermo Scientific UHPLC Ultimate 3000 apparatus (Thermo Fisher Scientific, Waltham, MA, USA), connected to an ESI-qTOF Compact (Bruker Daltonics, Bremen, Germany) HRMS detector was used. Separation was achieved on a Kinetex RP-18 column (100 × 2.1 mm × 2.6 μm; Phenomenex, Torrance, CA, USA). The UHPLC-ESI-MS instrument was operated in negative mode.

The instrument was calibrated with sodium formate cluster ions. The analyses of the obtained mass spectra were carried out using Data Analysis (Bruker Daltonics) software. The main instrumental parameters were as follows: scan range: 50–2200 *m/z*; dry gas: nitrogen; temperature: 200 °C; potential between the spray needle and the orifice: 4.2 kV. The gradient elution system consisted of 0.1% HCOOH in water (mobile phase A) and 0.1% HCOOH in acetonitrile (mobile phase B). At the flow rate of 0.3 mL/min, the following elution program was used: 0→30 min (5→95% B), 30→40 min (95% B), 40→45 min (95→5% B), 45→50 min (5% B). All analyses were carried out isothermally at 30 °C. The injection volume was 3 μL.

### 2.5. Quantitative Analysis

The total phenolic content of the methanol extract from *S. virgaurea* herbs was determined using the Folin–Ciocâlteu method, as described previously by Chandra et al. [32] with slight modifications. The content of total phenolics was expressed as mg gallic acid equivalents GAE/g of dry plant extract. The evaluation of the total flavonoid content of the extract from *S. virgaurea* herbs was performed using a spectrophotometric method [32]. The flavonoid content was expressed as mg quercetin equivalents QE/g of dry plant extract. The absorbance of the reaction mixtures was measured with a Thermo Scientific Multiskan GO UV/VIS spectrophotometer. The experiments were performed in triplicate.

### 2.6. Minimum Inhibitory Concentration (MIC) Determination

The minimum inhibitory concentrations (MICs) of amikacin, ciprofloxacin and *S. virgaurea* methanol extract were determined using the broth microdilution method, according to the Clinical and Laboratory Standards Institute guidelines (CLSI, 2020) [33].

### 2.7. Effect of S. virgaurea Extract, Antibiotics and Their Combinations on Bacterial Survival

The bacteria were grown overnight on nutrient agar (Biomed, Poland) and then transferred to MHB and incubated at 37 °C for 30 min. Following incubation, the bacterial culture was centrifuged (4000 rpm; 20 min) and suspended in phosphate-buffered saline (PBS) to obtain the final density of 0.5 in the McFarland scale. Bacterial suspension and antimicrobial agents were mixed together to obtain proper sub-inhibitory concentrations in each sample (0.25× MIC for *S. virgaurea* extract and 0.5× MIC for antibiotics). Samples were incubated at 37 °C for 24 h, then diluted and cultured on nutrient agar plates. After 0, 1, 3, 5, and 24 h of incubation at 37 °C the number of CFU/mL was counted. Control samples in all experiments contained no antibiotics or plant extract.

### 2.8. Effect of S. virgaurea Extract, Antibiotics and Their Combinations on Biofilm Production

The biofilm production assay was performed according to [34] with slight modifications, as described previously [35]. Samples were prepared in microtiter plate wells by adding the appropriate volume of *S. virgaurea* extract, amikacin, ciprofloxacin, *S. virgaurea* extract with amikacin and *S. virgaurea* extract with ciprofloxacin to 200 μL of MHB containing 20 μL of culture of bacteria (0.5 in the McFarland scale). On the basis of ODs of bacterial biofilm, the bacteria were classified as follows: OD ≤ ODc: no biofilm producer; ODc < OD ≤ 2× ODc: weak biofilm producer; 2× ODc < OD ≤ 4× ODc: moderate biofilm producer; OD > 4× ODc: strong biofilm producer. In our study, the ODc value was 0.197. The experiment was repeated three times for each growth condition and the results were averaged.

### 2.9. Postantibiotic Effect (PAE) and Postantibiotic Sub-MIC Effect (PASME) in the Presence of S. virgaurea Extract

After incubation for 2 h, bacteria were diluted to obtain an inoculum of approximately 5 × 10^7^ CFU/mL at the beginning of the experiment. To determine the PAE values, the strain was exposed to 1× MIC of amikacin, 1× MIC ciprofloxacin, 1× MIC of amikacin with 0.25× MIC of *S. virgaurea* extract, and 1× MIC of ciprofloxacin with 0.25× MIC of *S. virgaurea* extract. To determine the PASME values, the strain was exposed to 0.5× MIC of amikacin, 0.5× MIC ciprofloxacin, 0.5× MIC of amikacin with 0.25× MIC of *S. virgaurea* extract, and 0.5× MIC of ciprofloxacin with 0.25× MIC of *S. virgaurea* extract. The sample containing unexposed bacteria was the control. All samples were shaken aerobically in a water bath at 37 °C for 1 h. The antimicrobial agents were washed out using PBS and centrifugation (4000× *g* for 5 min). The control culture was also centrifuged. The treated and control cultures were placed in fresh MHB and were incubated in a water bath at 37 °C for 24 h. Growth was monitored hourly for 7 h and after 24 h by removing 100-µL samples, performing serial dilutions, and determining the number of CFU of the sample per mL on agar plates [36].

The PAE/PASME was defined as PAE/PASME=T−C, where T is the time required for the count in the test culture to increase 1 log10 above the count observed immediately after the removal of the antibacterial agent (time T_0_), and C is the time required for the count in the untreated count to increase 1 log10 above the count observed at time T_0_.

### 2.10. Statistical Analysis

The results are given as a mean value from three separate experiments. All values are expressed as a mean ± SD. Statistical differences between bacteria exposed to antibiotics, *S. virgaurea* extract and unexposed bacteria were analyzed using the non-parametric Kruskal–Wallis test, followed by Dunn’s multiple comparison test. Statistical calculations were made using STATISTICA 13.1 (Stat Soft, Krakow, Poland). *p* values ≤ 0.05 were considered to be statistically significant.

## 3. Results

### 3.1. Qualitative and Quantitative Analysis of S. virgaurea Extract

Tentative identification of the main compounds recorded by LC-DAD was performed on the basis of their accurate masses using high-resolution mass spectrometry. The mass accuracy for all separated compounds was within 2 ppm. The identified compounds and their MS data are shown in Figure 1 and Table 1. The main peaks on the UHPLC-chromatogram belong to chlorogenic acid, rutin and caffeoylquinic acid isomers. These compounds were previously described in *S. virgaurea* and some other *Solidago* species [2,37]. However, we were unable to characterize the compound corresponding to peak 8 at *m/z* 349.0934. The total phenolic and total flavonoid content of the goldenrod methanol extract were determined and they were 50.83 mg GAE/g dry weight and 33.15 QE/g dry weight, respectively.

### 3.2. Minimum Inhibitory Concentrations (MICs) of Antibiotics and S. virgaurea Extract

The MICs of amikacin, ciprofloxacin and *S. virgaurea* extract were 1.0 μg/mL, 0.007 μg/mL and 45 mg/mL, respectively.

### 3.3. Effect of S. virgaurea Extract, Antibiotics and Their Combinations on Bacterial Survival

Analysis of the data presented in Figure 2 shows that *S. virgaurea* extract exhibits antibacterial effects compared to the control sample. However, the strongest and most effective anti-growth effect of the goldenrod extract was found in young 1- and 3-h bacterial cultures (*p* ≤ 0.05). The hypothesis of our research assumed that the extract would show a synergistic effect with the antibiotics amikacin and ciprofloxacin, commonly used in the treatment of UTIs. However, the research results do not support this assumption. It can be noted that *E. coli* showed better growth in mixtures containing both *S. virgaurea* extract and antibiotics than in the samples in which bacteria were incubated with the drug itself. This situation was observed in T_5_ and T_24_ cultures (*p* ≤ 0.05). Thus, the subinhibitory concentration of the goldenrod extract (0.25× MIC) does not support the activity of ciprofloxacin and amikacin. On the contrary, it shows a protective effect against *E. coli* treated with antibiotics.

### 3.4. Effect of *S. virgaurea* Extract, Antibiotics and Their Combinations on Biofilm Production

OD values of control samples at all measurement times (T_24_, T_48_, T_72_) were in the range of 0.197 < OD ≤ 0.394 (Figure 3), which indicates weak biofilm mass formation by *E. coli* rods. OD values lower than 0.197, indicating the lack of biofilm formation, were observed only in 24-h biofilms treated with ciprofloxacin and amikacin at 0.5× MIC (*p* ≤ 0.05). Analysis of the data in Figure 3 (24 h) also shows that there was a decrease in biofilm formation for the samples containing the goldenrod extract, as well as its mixtures with antibiotics. In 48-h biofilm cultures, no statistically significant decrease in biofilm-mass formation was noted (*p* > 0.05). Therefore, it can be concluded that neither the subinhibitory concentration of *S. virgaurea* extract (0.25× MIC) nor combinations of drugs with plant extract inhibited the formation of biofilm mass by the *E. coli* strain. It is also noteworthy that in 72-h cultures containing a mixture of subinhibitory levels of drugs and the goldenrod extract, biofilm formation by *E. coli* strain was higher than in the control sample (*p* ≤ 0.05). It seems that the goldenrod extract at 0.25× MIC, when added to biofilm cultures with antibiotics, not only shows protective properties (protecting enterobacteria from drugs) but even stimulates bacteria to form a biofilm.

### 3.5. Postantibiotic Effect (PAE) and Postantibiotic Sub-MIC Effect (PASME) in the Presence of S. virgaurea Extract

The duration of PAE of ciprofloxacin, amikacin and *S. virgaurea* extract, as well as their mixtures (antibiotic + extract), against the *E. coli* strain was determined. The results pertaining to the post-antibiotic effects using ciprofloxacin at 1× MIC, amikacin at 1× MIC and plant extract at 0.25× MIC are presented in Figure 4 and Figure 5. The above-mentioned duration was determined according to the formula: PAE = T−C, where T is the time required for bacteria treated with the drug to increase their number by one logarithmic magnitude (1 log10) above the value found immediately after elimination of the drug (at T_0_) and C is the time necessary for the control culture bacteria to increase their number by one logarithmic magnitude compared to T_0_. T and C values derived from the charts in Figure 4 and Figure 5 are presented in Table 2, which also includes PAE values.

The goldenrod extract, as well as the one combined with ciprofloxacin at 1 × MIC, showed a short-term post-antibiotic effect (PAE), which lasted only 45 min (Figure 4, Table 2). A slightly longer duration of PAE (1 h) was observed in the presence of the extract and amikacin (Figure 5, Table 2). *E. coli* bacteria incubated only in the presence of ciprofloxacin increased their growth by one logarithmic value after 2 h. Amikacin showed the longest duration of PAE, lasting 3 h 30 min (Table 2).

Subsequently, the duration of the post-antibiotic sub-MIC effect (PASME) of ciprofloxacin, amikacin and *S. virgaurea* extract, as well as their combination (antibiotic + extract), against the *E. coli* strain was determined. The results of this experiment, using ciprofloxacin at 0.5× MIC, amikacin at 0.5× MIC and goldenrod extract at 0.25× MIC, are presented in Figure 6 and Figure 7. The duration of PASME was determined according to the same formula as the one of PAE. T and C values derived from the charts in Figure 6 and Figure 7 are presented in Table 3, which also includes PASME values.

The goldenrod extract combined with ciprofloxacin and the extract combined with amikacin showed a short-term PASME which lasted only 30 min. *E. coli* bacteria treated with the extract itself increased their growth by one logarithmic value after 45 min. Ciprofloxacin and amikacin showed the longest duration of PASME, lasting 1 h 45 min and 2 h 15 min, respectively.

Comparing PAE and PASME results, it can be concluded that the duration of PASME was always shorter than PAE in samples containing antibiotics, as well as in samples containing antimicrobial agents and plant extract.

## 4. Discussion

*S. virgaurea* is a plant that has been used in traditional medicine for centuries as an agent in the prevention and treatment of various diseases [19,20]. However, the number of reports describing the antibacterial effect of the goldenrod extracts is very limited [29,30,31]. Thiem and Goślińska [29] analyzed the bactericidal properties of methanolic and ethanolic extracts of goldenrod herbs against *E. coli*, *Bacillus subtilis*, *Bacillus pumilis*, *Proteus mirabilis*, *Proteus vulgaris*, *Micrococcus luteus*, *Pseudomonas aeruginosa*, *Staphylococcus aureus* and *Staphylococcus epidermidis*. The authors established the MBC values (the minimum bactericidal concentration) of goldenrod extracts using the diffusion method. In the case of *E. coli* rods, the MBC value of the ethanol extract was 31.2 mg/mL, whereas that of the methanol extract was 62.5 mg/mL. Demir et al. [30] studied the antibacterial activity of *S. virgaurea* methanol extract against *E. coli* ATCC 35218, *E. coli* ATCC 25922, *S. aureus* ATCC 25923, *P. aeruginosa* ATCC 27853, *B. subtilis* ATCC 29213, *Bacillus cereus* NRLL B-3008, *Enterobacter feacalis* ATCC 292112 and five clinical *Klebsiella pnemoniae* isolates using the disc diffusion method. The growth inhibition zones were present only for *E. coli* ATCC 25922, *S. aureus*, *B. cereus* and *E. feacalis*. Anžlovar and Koce [31] determined the antibacterial activities of four different *S. virgaurea* extracts (acetone, methanolic, ethanolic and aqueous) against *E. coli*, *S. aureus* and *B. subtilis*. None of the extracts limited the growth of the tested bacterial strains. The above-described studies, although interesting, are limited to determining only the values of MBC and the zone of inhibition of bacterial growth. In our study, apart from determining the MIC value, the effect of the extract of *S. virgaurea*, as well as its combination with antibiotics, on the survival of bacteria, biofilm formation and the duration of the post-antibiotic effect of amikacin and ciprofloxacin was also determined. In the current study, the MIC value of goldenrod methanol extract was established, which was 45 mg/mL. The extract decreased the number of bacterial cells in planktonic cultures and limited the biofilm-mass formation in young biofilm.

The dynamic development of antibiotic treatment greatly facilitated the fight against bacterial infections. Unfortunately, the widespread abuse and misuse of antibiotics have led to an increase in the drug resistance of many bacterial strains. Therefore, the role of herbal medicine as an alternative to antibiotic therapy is increasingly being raised. Research on already-known plant products is now being conducted on a large scale and several phytotherapeutics have been found. Essential oils (EOs) are one of the subjects of research for use as an alternative in combating pathogenic bacteria for humans. EOs contain terpenes such as carvacrol, menthol, geraniol and thymol, possessing antimicrobial properties [38]. Moustafa et al. [39] investigated the antibacterial potential of different EOs against multidrug-resistant UPECs. The obtained results revealed that clove and oregano EOs were the most active agents compared with other assessed EOs. Nabti et al. [40] determined the antibacterial activity of *Origanum glandulosum* EOs against multidrug-resistant UPECs. EOs were active against all the tested strains, showing similar inhibition zone diameters, as well as MIC and MBC values. Le et al. [41] investigated the bioactivities of the essential oil of *Atalantia sessiflora* against drug-resistant *E. coli* ATCC 35218, *P. aeruginosa* ATCC 27853, *S. aureus* ATCC 43300 and different clinical strains (S*. aureus*, *E. faecalis*, *K. pneumoniae*, *P. aeruginosa*). The oil showed antimicrobial activities against *S. aureus*, *K. pneumoniae* and *E. coli* strains. Ebani et al. [42] investigated the antibacterial activity of five EOs against multi-drug resistant UPECs. *Illicium verum* and *Salvia sclarea* oils exhibited moderate growth inhibition against *E. coli* strains. Better results have been obtained for *Origanum vulgare*, *Thymus vulgaris* and *Ocimum basilicum* oils.

The increasing number of bacteria that are resistant to antibiotics has contributed to the failure of the treatment of infections caused by them and is a serious health problem in medicine worldwide. Therefore, it is reasonable to conduct research on the effect of plant extracts and essential oils to enhance the antimicrobial activity of antibiotics. Some researchers combine antibiotics with plant compounds, e.g., EOs, to determine transcriptional changes in the bacterial genome, receiving information on key genes that underlie the synergism between essential oils and antibiotics. Lai et al. [24] conducted transcriptomic analysis of multidrug-resistant *E. coli* to identify any presence of transcriptional changes in the genome upon the use of a combination of lavender essential oil (*Lavandula angustifolia*) (LEO) and piperacillin. Analysis of the results revealed that the use of LEO-piperacillin caused the upregulation of certain genes which affected fructose and mannose metabolism, the metabolism of *E. coli* in diverse environments and nitrotoluene degradation, as well as the phosphotransferase system. The combinatory treatment also caused the downregulation of the *aldA* gene, for example, which is crucial for the survival and replication of *E. coli* strains under harsh conditions. The above results show that the presence of LEO can increase the potential of piperacillin for cellular destruction. Some research has shown that the presence of plant components enhances the potency of antibiotics by weakening bacteria’s ability to develop resistance. Scazzocchio et al. [25] showed that the presence of coriander essential oil (*Coriandrum sativum*) reduced the MIC of gentamicin against multidrug-resistant UPECs. A strong synergy was reported for two Moroccan thyme (*Thymus maroccanus* and *T. broussonetii*) EOs in combination with chloramphenicol against two strains: wild-type *E. coli* K-12 (AG100) and *E. coli* K-12 mutant (AG102) overexpressing the AcrAB pump, responsible for resistance to tetracycline, chloramphenicol, ampicillin, nalidixic acid and rifampicin. Both EOs decreased the MIC values of chloramphenicol against *E. coli* AG100 and *E. coli* AG102 strains by 4- and 32-fold, respectively [43,44]. Similar results were obtained for another Moroccan thyme (*Thymus riatarum*) EO in combination with chloramphenicol against *E. coli* K-12 (AG100) and *E. coli* K-12 mutant (AG100A) with the AcrAB pump deleted. *T. riatarum* EO decreased the MIC values of chloramphenicol against AG100 and AG100A by 4- and 2-fold, respectively [26]. The results suggest that the synergy could be attributed to efflux pump activity changes by thyme EOs, thus increasing the activity of chloramphenicol. Duran et al. [45] examined the effects of *Nigella sativa* EOs against drug-resistant *E. coli* strains. Following treatment with essential oils, resistance rates of *E. coli* strains against ampicillin, levofloxacin and cefuroxime decreased—the growth inhibition zone diameters of *E. coli* strains were higher and the MIC values of the antibiotics were lower when compared to bacteria not treated with the EOs. In the research conducted by Miladinovic et al. [27] lemon thyme (*Thymus pulegioides*) EO showed synergistic effects with tetracycline and chloramphenicol against *E. coli* ATCC 25922. It is worth noting that the combination of lemon thyme EO and streptomycin displayed an antagonistic effect against *E. coli* ATCC 25922. This phenomenon is probably caused by competition for the same target, with the bacterial 30S ribosomal subunit being crucial in protein synthesis.

Despite the various studies describing the antibacterial activities of the combinations of antibiotics and plant materials such as those mentioned above [24,25,26,27,43,44,45], research on the interactions between *S. virgaurea* extract and antibiotics is not to be found. Therefore, we consider our research and the results obtained to be original and innovative, expanding the knowledge of the effects of antibiotics and *S. virgaurea*. One of the main objectives of the research presented in this paper was to determine the effect of both subinhibitory concentrations of the European goldenrod herb extract and mixtures of this extract with subinhibitory concentrations of ciprofloxacin and amikacin on planktonic and biofilm cultures of uropathogenic *E. coli*. The research hypothesis assumed the synergistic effect of the goldenrod herb extract and both antibiotics. However, the results showed that the methanol extract from *S. virgaurea* at a concentration of 0.25× MIC does not support the activity of the drugs against *E. coli* strain. This extract exhibited an antagonistic effect against ciprofloxacin and amikacin, which manifested itself in their weaker activity against both planktonic bacteria and biofilm bacteria. The available literature indicates that the causes of this phenomenon should be sought in the chemical composition and mechanisms of activity of the extract. For instance, the goldenrod extract contains polyphenols, of which the effects are very various. The activity of certain groups of bacteria is decreased by some polyphenols, whereas other species are stimulated to grow by the same compounds. Tzounis et al. [46] showed that catechin weakened the growth of *Clostridium histolyticum* and increased the growth of *E. coli*, the *Eubacterium rectale*–*Clostridium coccoides* group, *Lactobacillus* spp. and *Bifidobacterium* spp. The growth of *Bifidobacterium* and *Lactobacillus* was also enhanced by resveratrol, pomegranate ellagitannins and urolitin A [47,48]. In rat studies, tannins weakened the growth of *Clostridium leptum*, while stimulating the growth of *Bacteroides* [49].

There are numerous data on the antibacterial properties of plant polyphenols. The therapeutic (including antibacterial) properties of propolis, which contains large amounts of flavonoids, are widely known [50]. A group of polyphenols which is particularly common in red and purple fruit (cherries, raspberries, blueberries), showing bacteriostatic and bactericidal effects against many pathogens such as *Staphylococcus*, *Klebsiella*, *Helicobacter* and *Bacillus* is anthocyanins [51]. In the study on the antibacterial properties of the extract from *Solidago chilensis* rhizomes, its growth inhibitory effect on *E. coli*, *Pseudomonas aeruginosa* and *Staphylococcus aureus* strains was observed. In this extract, caffeic and chlorogenic acids are present [52]. Polyphenolic compounds, due to the presence of various chemical groups, including hydroxyl residues, tend to be incorporated into lipid membranes, causing changes in their fluidity and permeability. This increases the susceptibility of pathogens to antibacterial agents and may generate the outflux of cytoplasmic substances which are essential for the microbe, thus inhibiting the growth of the microorganism [49,52]. Studies using plant extracts and polyphenols isolated from them confirm that these compounds may show synergism with each other and with other antimicrobial agents in terms of a biocidal/biostatic activity. In many cases, stronger effects of mixtures of plant compounds than their individual components are observed, as well as an increase in the activity of antibiotics when they are administered together with these compounds. It has been demonstrated that quercetin increases the permeability of bacterial cell membranes, which may increase their susceptibility to antibiotics even in the case of pathogens showing drug resistance [50,53]. Quave et al. [54] showed that a mixture of compounds derived from blackberries (*Rubus ulmifolius*) which contain ellagic acid (among other components), intensified the effects of daptomycin, clindamycin and oxacillin against a methicillin-sensitive *S. aureus* (MSSA) strain. They also observed a significant inhibition of biofilm formation by both the MSSA strain and methicillin-resistant *S. aureus* (MRSA) in the presence of blackberry extract [54]. The biofilm forms show much greater resistance to antibiotics than the planktonic forms. The inclusion of natural plant compounds in biofilm structure weakens the bacterial adhesion to the invaded surface, which in turn reduces the formation of biofilm mass. In this way, the synergism of the activity of drugs and plant metabolites can be explained.

Scientists have proven that during antibiotic therapy it is possible to reduce the toxic dose of a drug by extending the time between successive doses administered to the patient [28]. Therefore, the determination of PAE and PASME is a good basis for planning the treatment with some chemotherapeutic agents. Hence, our research aimed to determine the PAE and PASME of ciprofloxacin and amikacin in the presence of the goldenrod herb extract. The research hypothesis assumed that the duration of PAE and PASME of ciprofloxacin and amikacin would be extended in the presence of goldenrod herb extract. However, our results indicated that preincubation of *E. coli* with drugs and *S. virgaurea* extracts shortened the duration of both PAE and PASME. Therefore, the hypothesis posed at the beginning of the study proved to be false. The study results differed from these obtained by D’Arrigo et al. [36], who determined the duration of PAE and PASME of tobramycin combined with tea tree oil against gram-positive *S. aureus* and gram-negative *E. coli* bacteria. Tobramycin at concentrations of 1× MIC and 2× MIC was used in the study. The duration of PAE against *E. coli* rods was longer when using tobramycin at 1× MIC and lasted 1 h 20 min. This time increased to 10 h 50 min when *E. coli* rods were short-term treated with a mixture of tobramycin (1× MIC) and tea tree oil (0.05%). A similar dependence was observed when using antibiotics in subinhibitory concentrations of 0.1, 0.2 and 0.3× MIC, in which the duration of PASME was 2 h 40 min, 3 h 20 min and 5 h 20 min, respectively. After plant oil addition, the above-mentioned duration of PASME was extended to 13 h 40 min, 14 h 30 min and 15 h, respectively. The duration of the PAE of tobramycin against *S. aureus* at 1× MIC was 1 h 40 min and it was extended to 10 h 25 min after tea tree oil (0.25%) was added to tobramycin. Interestingly, after 0.5% plant oil was added, the duration of PAE was extended to 17 h 25 min. After 0.5% tea tree oil was added to subinhibitory concentrations of tobramycin, the duration of PASME was twice as long as with the use of 0.25% oil. The above-mentioned results show that tea tree oil extended the growth inhibition time of both *E. coli* and *S. aureus*. Aboulmagd et al. [55] also studied the PAE using imipenem and *Camellia sinensis* green tea leaf extract against the MRSA strain. In the experiment, the plant extract at 0.25× MIC and the antibiotic at 1, 2 and 3× MIC were used. The duration of PAE was imipenem-concentration-dependent and ranged from 1 h 30 min to 2 h 50 min. After the green tea leaf extract was added, PAE was extended to 2 h 55 min–4 h 40 min. Braga et al. [56] conducted an experiment on MRSA and MSSA clinical strains. In their study, the effect of a *Punica pomegranate* methanolic extract on the duration of PAE of ampicillin was determined. PAE was extended from 3 to 7 h. Hussin et al. [57] examined the duration of the PAE of tetracycline combined with extracts from pomegranate, thyme, myrrh balm and cornelian against *E. coli*. The presence of extracts from pomegranate, myrrh balm and thyme significantly increased the duration of PAE of tetracycline to 3, 4 and 5 h, respectively. The results obtained by the above-mentioned scientists differ significantly from those presented in the current report. Those papers [36,55,56,57] present synergism in the interaction of plant extracts with antibiotics, expressed in the extension of the growth inhibition time of bacteria. In the current authors’ research, however, it has been shown that the use of the goldenrod herb extract shortens the duration of the PAE and PASME of both ciprofloxacin and amikacin.

To summarize the research presented here, it can be stated that the goldenrod herb extract limited the growth of the planktonic forms of UPEC at a very early stage of culture and decreased the production of 24-h biofilm. However, the addition of the extract to amikacin and ciprofloxacin did not enhance their antibacterial activity, nor did it extend the duration of PAE and PASME. On the contrary, in the cultures treated with both the extract and antibiotic, the number of *E. coli* bacteria increased. The amount of biofilm mass they formed also increased and the post-antibiotic effect became shorter. The results of our research indicate that the plant extract does not always show synergistic and bacteriostatic activity with antibiotics. The causes of this phenomenon may include (i) ’competition’ between molecules of plant origin and antibiotic molecules for a specific target site in a bacterial cell or blocking of the capture region for these molecules; (ii) direct interactions between plant substances and antibiotics, leading to a reduction in drug activity; (iii) the ability of bacteria to transform active forms of plant metabolites into less active ones that cannot support drug activity. To understand the exact mechanisms responsible for these interactions, further and more detailed research is needed.

## 5. Conclusions

The results of the current preliminary research are novel and important because they show that the goldenrod herb extract limits the growth of the planktonic forms of UPEC and the formation of biofilm mass at early stages of culture. It shows no synergistic effect in combination with antibiotics and shortens the duration of the post-antibiotic effect of both amikacin and ciprofloxacin. When deciding to use a combination of *S. virgaurea* extract and amikacin/ciprofloxacin, it is necessary to take into account their antagonistic activity.

## Figures and Tables

**Figure 1 pharmaceutics-13-00573-f001:**
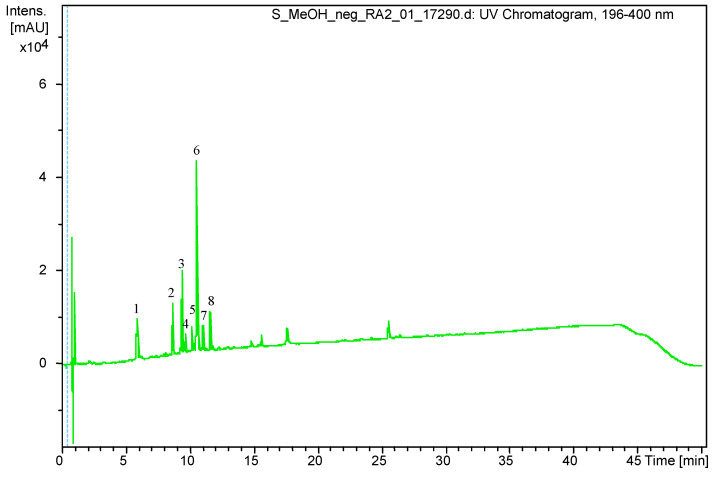
UHPLC-DAD chromatogram of the methanol extract of *S. virgaurea* obtained at 196–400 nm.

**Figure 2 pharmaceutics-13-00573-f002:**
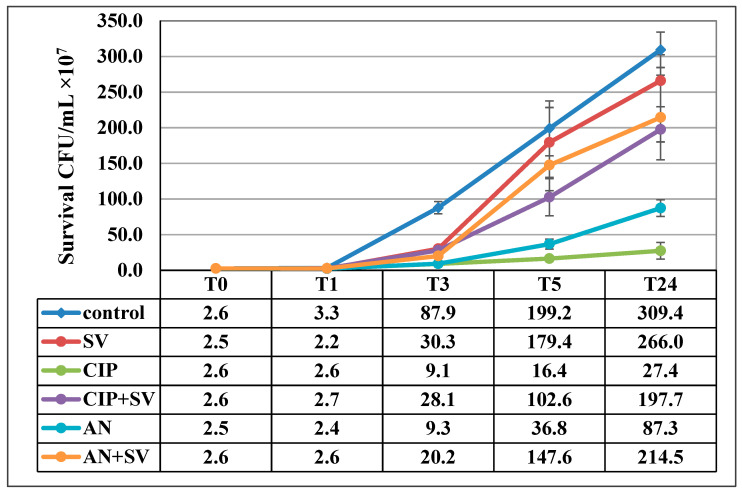
Survival of *E. coli* in the presence of *S. virgaurea* extract (SV), ciprofloxacin (CIP), amikacin (AN), *S. virgaurea* extract with ciprofloxacin (CIP+SV) and *S. virgaurea* extract with amikacin (AN+SV). Values represent the mean ± SD of three separate experiments.

**Figure 3 pharmaceutics-13-00573-f003:**
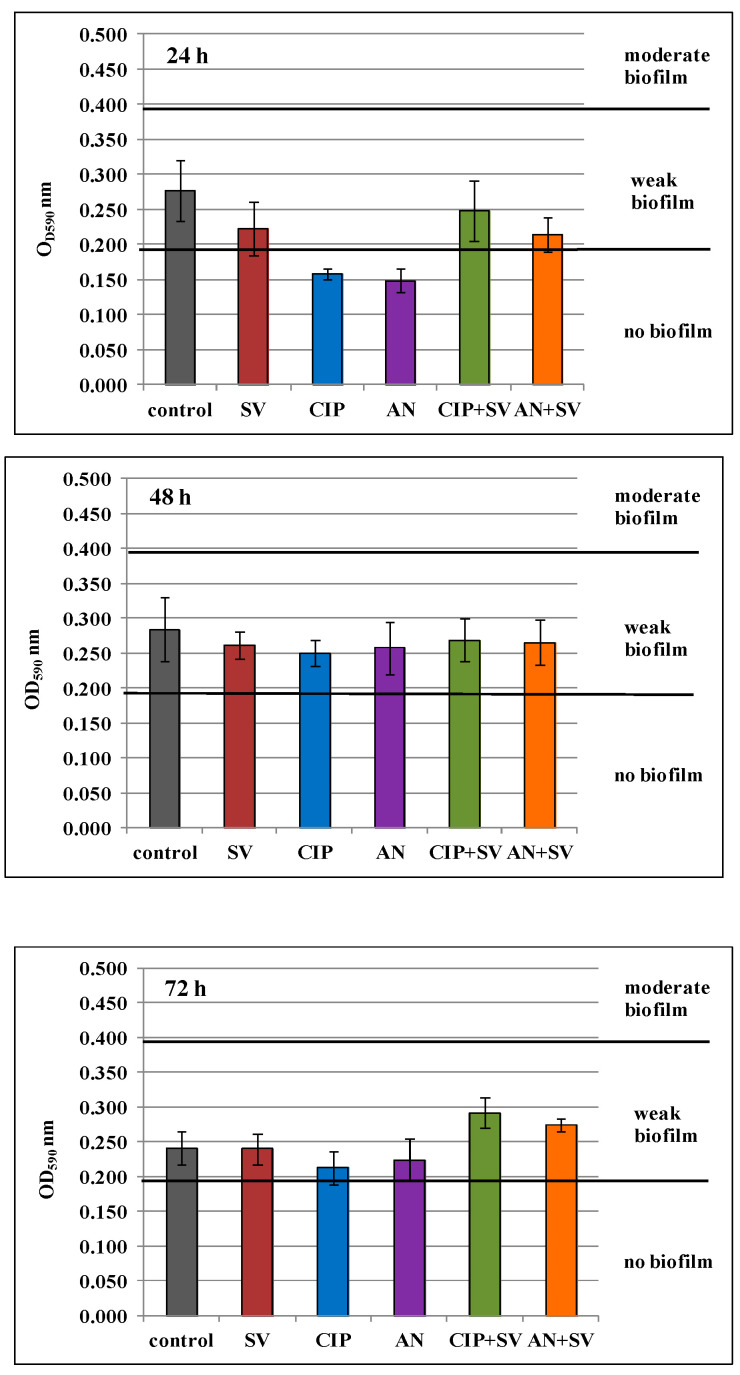
Biofilm formation in the presence of *S. virgaurea* extract (SV), ciprofloxacin (CIP), amikacin (AN), *S. virgaurea* extract with ciprofloxacin (CIP+SV) and *S. virgaurea* extract with amikacin (AN+SV). Values represent the mean ± SD of three separate experiments.

**Figure 4 pharmaceutics-13-00573-f004:**
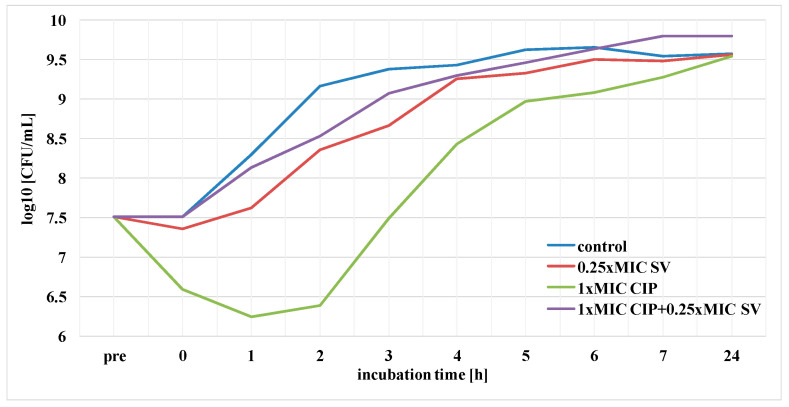
PAE of ciprofloxacin (CIP) against *E. coli* in the presence of *S. virgaurea* extract (SV).

**Figure 5 pharmaceutics-13-00573-f005:**
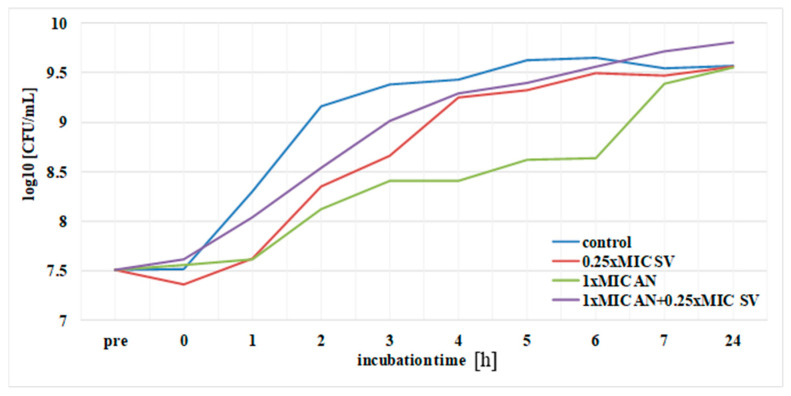
PAE of amikacin (AN) against *E. coli* in the presence of *S. virgaurea* extract (SV).

**Figure 6 pharmaceutics-13-00573-f006:**
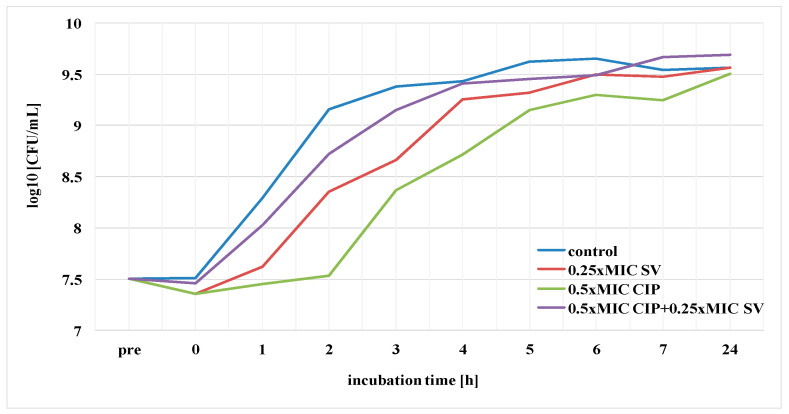
PASME of ciprofloxacin (CIP) against *E. coli* in the presence of *S. virgaurea* extract (SV).

**Figure 7 pharmaceutics-13-00573-f007:**
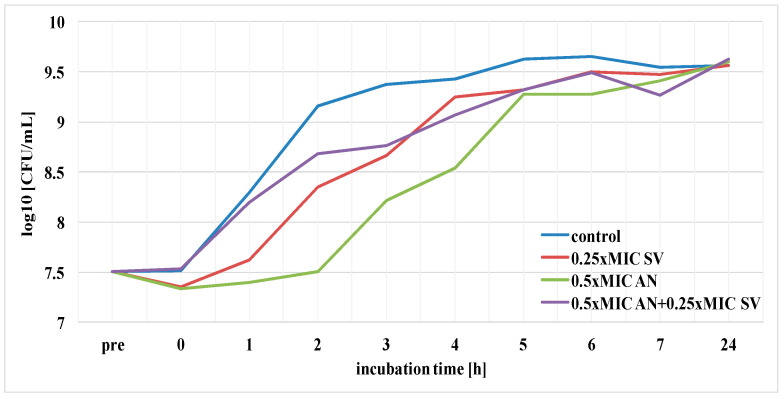
PASME of amikacin (AN) against *E. coli* in the presence of *S. virgaurea* extract (SV).

**Table 1 pharmaceutics-13-00573-t001:** Tentative assignment of UHPLC chromatogram main peaks as phenolic compounds, based on MS.

Peak Number.	Retention Time (R_t_)	(M – H) (*m/z*)	Identification
1	5.9	353.0878	Chlorogenic acid
2	8.6	613.1767	Leiocarposide
3	9.4	609.1460	Rutin
4	9.7	463.0881	Hyperoside/Isoquercitrin
5	10.2	593.1502	Kaempferol 3-robinobioside/ Kaempferol 3-rutinoside
6	10.6	515.1192	Dicaffeoylquinic acid isomer
7	11.1	515.1185	Dicaffeoylquinic acid isomer
8	11.6	349.0934	Unknown

**Table 2 pharmaceutics-13-00573-t002:** PAE of ciprofloxacin (CIP) and amikacin (AN) against *E. coli* in the presence S*. virgaurea* extract (SV).

Sample	T (h)	C (h)	PAE = T−C (h)
0.25× MIC SV	2 h	1 h 15 min	45 min
1× MIC CIP	3 h 15 min	1 h 15 min	2 h
1× MIC CIP + 0.25× MIC SV	2 h	1 h 15 min	45 min
1× MIC AN	4 h 45 min	1 h 15 min	3 h 30 min
1× MIC AN + 0.25× MIC SV	2 h 15 min	1 h 15 min	1 h

**Table 3 pharmaceutics-13-00573-t003:** PASME of ciprofloxacin (CIP) and amikacin (AN) against *E. coli* in the presence *S. virgaurea* extract (SV).

Sample	T (h)	C (h)	PASME = T−C (h)
0.25× MIC SV	2 h	1 h 15 min	45 min
0.5× MIC CIP	3 h	1 h 15 min	1 h 45 min
0.5× MIC CIP + 0.25× MIC SV	1 h 45 min	1 h 15 min	30 min
0.5× MIC AN	3 h 30 min	1 h 15 min	2 h 15 min
0.5× MIC AN + 0.25× MIC SV	1 h 45 min	1 h 15 min	30 min

## Data Availability

Not applicable.

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
