# Peer review of "Is it Worth Combining Solidago virgaurea Extract and Antibiotics against Uropathogenic Escherichia coli rods? An In Vitro Model Study"

_pharmaceutics, 2021, doi:10.3390/pharmaceutics13040573_

Round 1

Reviewer 1 Report

Dear authors

The plant extract reduced 24 hrs-formed biofilm. However, the combination of S. virgaurea extract with antibiotics decreased their antibacterial activity and shortened the duration of PAE and PASME. S. virgaurea extract demonstrated antibacterial activity against E. coli rods. However, its combination with antibiotics resulted in antagonistic interactions against bacteria. Therefore, the conclusion is that, contrary to common practice, S. virgaurea extract should not be combined with antibiotics, as it may cause therapeutic failure.

Analysis by paper partitions:   1 - Introduction: it must be reformed in the content and in the writing of the general part to review the syntax of the topic   3- Discussion: to deepen in consideration of the problem of antibiotic resistance the use of essential oils against multidrug-resistant strains, to add this study as a future perspective. Learn more about this by using and citing the following references: PMID: 33031096 ; PMID: 29987237 ; PMID: 32596098 3 - Check the bibliographic entries throughout the text, some of which are non-compliant, review some entries in the bibliographic references and necessarily insert those referred to in point 2 for the purpose of acceptance by me.   4 - Review the English grammar and in particular the applied scientific English: in particular, the verb tenses and the syntax in the discussion.

Author Response

Dear Reviewer,

we would like to thank you very much for all your suggestions, comments, and time spent reviewing our manuscript.

Answers:

1 - Introduction: it must be reformed in the content and in the writing of the general part to review the syntax of the topic  

According to your suggestions, the content of the Introduction section has been changed and reorganized. A detailed description of the composition of the Solidago virgaurea has been included. The role of plant compounds in increasing the activity of antibiotics against drug-resistant bacteria was emphasized. An attempt was also made to explain why our research is innovative compared to the results of other researchers published so far.

2- Discussion: to deepen in consideration of the problem of antibiotic resistance the use of essential oils against multidrug-resistant strains, to add this study as a future perspective. Learn more about this by using and citing the following references: PMID: 33031096; PMID: 29987237; PMID: 32596098; 

According to Reviewer’s suggestions, in the Discussion section we considered the problem of drug resistance in bacteria and the use of essential oils in combating these microorganisms. We introduced three suggested articles (PMID: 29987237; PMID: 32596098; PMID: 33031096) as well as several others (all the new references are red-coloured)                                          

3 - Check the bibliographic entries throughout the text, some of which are non-compliant, review some entries in the bibliographic references and necessarily insert those referred to in point 2 for the purpose of acceptance by me.  

We agree with Reviewer’s comment – we checked all the references in the text; we corrected citation errors and added original source references to the manuscript text.

4 - Review the English grammar and in particular the applied scientific English: in particular, the verb tenses and the syntax in the discussion.

The English language throughout the manuscript has been checked and corrected.

Reviewer 2 Report

Dear Authors,

The present study aims to offer scientific arguments for the possible use in therapy of Solidago virgaurea extracts. The research subject is interesting and brings scientific important data in the field, especially as it deals with the study of a common species, that is evaluated for an effect lesser studied by other authors. Some changes of the manuscript should nevertheless be performed in order to improve its quality. The major first observation regards the title, which is not quite adequate for the content of your study – such a title can make the reader think at clinical study and in fact your study is only an experimental one. Please revise English throughout the whole manuscript. Please try to avoid conclusions that overpass the results of your study: e.g. line 16 – “it may cause therapeutic failure” – it’s to much as you cannot conclude that by only performing one study with some experiments. Following specific changes should also be performed:

 Abstact: Please revise the aims of your study (lines 11-16) and rephrase them in order to make proper sentences. Revise English (e.g. “it was decided” is not a proper English expression) – same observation remains valid for the expression of your study’s aim in the end of the introduction.

Introduction: This part should contain the description of similar studies in scientific literature, if they exist. Please describe and cite similar studies and make an introduction into context of your study. Please add data on the originality of your study by comparison with similar ones existing in literature. State differences. In this way, novelty and originality of the study should become clearer and must be added.

Materials and methods: If your methods are not novel, this part should contain references for all methods – if they are adaptation of existing methods, please cite them and state differences.

Results: In the parts where your results are similar to the ones that are already cited in scentific literature, please state the novelties you bring – maybe better results? (e.g. UHPLC-DAD and MIC determination). For the UHPLC-DAD section, a quantification of compounds should be performed in order to establish the amounts of compounds. In fact, in this quantification you may find the key of your study – amounts of different compounds are crucial to the development of biological activities.  

Discussions: Please add more references in this part, you state different gereral issues, especially in the first part (lines 276-294). Line 319: caffeic acid

Conclusions: Please change last sentence, as it is not a conclusion of your study.

All these suggested changes should be performed in order to bring further improvements to the manuscript. 

Author Response

Dear Reviewer,

we would like to thank you very much for all suggestions, comments, and time spent reviewing our manuscript.

  1. The major first observation regards the title, which is not quite adequate for the content of your study – such a title can make the reader think at clinical study and in fact your study is only an experimental one.

We agree with Reviewer’s suggestion. Our proposal for the new title is: Is it worth combining Solidago virgaurea extract and antibiotics against uropathogenic Escherichia coli rods? An in vitro model study.

  1. Please revise English throughout the whole manuscript. English language has been corrected carefully

The English language throughout the manuscript has been checked and corrected.

  1. Please try to avoid conclusions that overpass the results of your study: e.g. line 16 – “it may cause therapeutic failure” – it’s to much as you cannot conclude that by only performing one study with some experiments.

We agree. This phrase: “it may cause therapeutic failure” has been removed.

Following specific changes should also be performed:

Abstract

  1. Please revise the aims of your study (lines 11-16) and rephrase them in order to make proper sentences.

Thank you for this suggestion – we revised aims and reorganized the Abstract

  1. Revise English (e.g. “it was decided” is not a proper English expression) – same observation remains valid for the expression of your study’s aim in the end of the introduction.

We agree, the phrase “it was decided” is improper and has been changed

Introduction

  1. This part should contain the description of similar studies in scientific literature, if they exist. Please describe and cite similar studies and make an introduction into context of your Please add data on the originality of your study by comparison with similar ones existing in literature. State differences. In this way, novelty and originality of the study should become clearer and must be added.

Thank you for this suggestion. Similar studies have been introduced in the Introduction section and also described in more detail in the Discussion section. We hope that now it will be clear to the reader that our research is important and innovative. All the changes are red-coloured.

Materials and methods:

  1. If your methods are not novel, this part should contain references for all methods – if they are adaptation of existing methods, please cite them and state differences.

Methods described in 2.6 and 2.8 subsections have been shortened.

Results: 

  1. In the parts where your results are similar to the ones that are already cited in scentific literature, please state the novelties you bring – maybe better results? (e.g. UHPLC-DAD and MIC determination). For the UHPLC-DAD section, a quantification of compounds should be performed in order to establish the amounts of compounds. In fact, in this quantification you may find the key of your study – amounts of different compounds are crucial to the development of biological activities.  

Total phenolic content and total flavonoid content of the goldenrod methanol extract were determined by the spectrophotometric method described in subsection 2.5. The results are included in the manuscript in subsection 3.1.

Discussions

  1. Please add more references in this part, you state different general issues, especially in the first part (lines 276-294).

The Discussion section has been thoroughly reorganized and extended, and more references have been added to the manuscript. All the new references are red-coloured.

Line 319: caffeic acid – has been changed

Conclusions:

  1. Please change last sentence, as it is not a conclusion of your study.

The last sentence has been changed as suggested.

Round 2

Reviewer 2 Report

Dear Authors,

The present study aims to offer scientific arguments for the possible use in therapy of Solidago virgaurea extracts. After the first round of review, the manuscript is significantly improved, especially in the major concern, which is represented by the title. Authors should nevertheless perform some minor changes in the abstract. Please revise all abstract. The short version is too short and does not provide enough details about the study and it does not follow the structure of the manuscript. The long one is too long and contains more than 200 words, as suggested for authors and it should be shortened. In the short one you have some numbers which do not belong there. Please revise all these.

Author Response

Dear Reviewer,

The abstract has been corrected - the "short version" appeared in the manuscript by accident. It was a complete oversight for which we apologize. The current version of the abstract has been slightly modified and shortened. It currently contains 196 words.

Thank you again for your valuable recommendations and time spent revising our manuscript.

Best regards

This manuscript is a resubmission of an earlier submission. The following is a list of the peer review reports and author responses from that submission.